# scHyena: Foundation Model for Full-Length Single-Cell RNA-Seq Analysis in Brain

## Abstract

Single-cell RNA sequencing (scRNA-seq) has made significant strides in unraveling the intricate cellular diversity within complex tissues. This is particularly critical in the brain, presenting a greater diversity of cell types than other tissue types, to gain a deeper understanding of brain function within various cellular contexts. However, analyzing scRNA-seq data remains a challenge due to inherent measurement noise stemming from dropout events and the limited utilization of extensive gene expression information. In this work, we introduce scHyena, a foundation model designed to address these challenges and enhance the accuracy of scRNA-seq analysis in the brain. Specifically, inspired by the recent Hyena operator, we design a novel Transformer architecture called singe-cell Hyena (scHyena) that is equipped with a linear adaptor layer, the positional encoding via gene-embedding, and a bidirectional Hyena operator. This enables us to process full-length scRNA-seq data without losing any information from the raw data. In particular, our model learns generalizable features of cells and genes through pre-training scHyena using the full length of scRNA-seq data. We demonstrate the superior performance of scHyena compared to other benchmark methods in downstream tasks, including cell type classification and scRNA-seq imputation.

## 1 Introduction

Single-cell RNA sequencing (scRNA-seq) is a powerful technique for profiling gene expression levels at single-cell resolution, enabling molecular characteristics of complex biological systems in both normal and disease states (Saliba et al., 2014; Rood et al., 2022). Through scRNA-seq, several key objectives can be achieved, including cell type annotation (Li et al., 2020; Hao et al., 2021), the discovery of novel cell types (Villani et al., 2017), the identification of marker genes (Jaitin et al., 2014), and the analysis of cellular heterogeneity (Papalexi & Satija, 2018; Kinker et al., 2020). It is worth noting that the brain exhibits a particularly diverse range of cell types compared to other tissues (Saunders et al., 2018; Hodge et al., 2019). Therefore, conducting scRNA-seq analysis in the brain is especially important to gain a deeper understanding of brain function within various cellular contexts.

Despite its utility, there are significant challenges in scRNA-seq data analysis. Firstly, the quantity of mRNA in a single cell is quite limited so there is a risk of failing to capture gene expression, known as the 'dropout' phenomenon. Consequently, scRNA-seq data often contains numerous zero counts, and it becomes crucial to distinguish between true and false zero counts. To address dropout events, various imputation methods for scRNA-seq data have been developed (Huang et al., 2018; Li & Li, 2018; Van Dijk et al., 2018; Arisdakessian et al., 2019; Eraslan et al., 2019). However, it is worth noting that many of these existing methods tend to have long computational runtimes. Hence, there is a need for an efficient method to impute missing values in scRNA-seq data.

Another challenge arises from the long sequence length of scRNA-seq data. Typically, scRNA-seq measures the expression levels of tens of thousands of genes, and cell type annotation methods (Aran et al., 2019; Li et al., 2020; Hao et al., 2021; Yang et al., 2022b) classify cell types based on gene expression patterns. However, dealing with information from all genes can be challenging due to the high computational complexity requirements or the limited capacity of the models. As a result, many annotation methods select highly variable genes (HVGs) consisting of a few thousand genes and rely solely on the expression levels of these HVGs. However, the selection of HVGs is not only

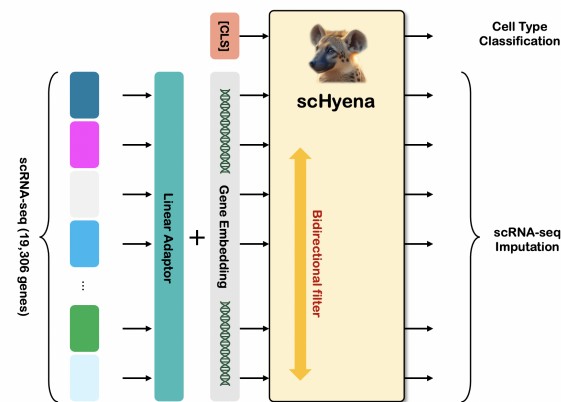

Figure 1: The proposed scHyena consists of several novel innovations: linear adaptor layer, gene-embedding, and bidirectional Hyena operator. Following pre-training, scHyena can be applied to various downstream tasks, including cell type classification and scRNA-seq imputation.

sensitive to parameter choices but also subject to variability across different datasets and batches (e.g., different patients). Furthermore, if the number of HVGs chosen is insufficient, it may lead to the loss of important cellular information. Therefore, there is a need for analysis methods capable of handling information from all genes.

In recent years, the foundation models have gained attention and have been explored for a wide range of data types (Devlin et al., 2018; Brown et al., 2020; Bommasani et al., 2021; Ramesh et al., 2021). In general, foundation models undergo a pre-training phase using unlabeled, extensive datasets via self-supervised learning to acquire generalizable features within the data domain. Following this pre-training, these foundation models can be effectively applied to various downstream tasks by fine-tuning them with smaller, labeled datasets. More recently, a novel operator known as Hyena (Poli et al., 2023) has been introduced as an alternative to the self-attention mechanism in Transformers to reduce the computational complexity of self-attention. Thanks to this reduction in complexity, Hyena is capable of handling input sequences containing hundreds of thousands of tokens, leading to improved performance when dealing with long sequences.

Inspired by these developments, here we introduce scHyena, a foundation model that incorporates the Hyena operator for the analysis of scRNA-seq data from brain tissues (Fig. 1). scHyena is a Transformer-based model equipped with the Hyena operator, enabling it to process scRNA-seq data without the need for dimension reduction or the selection of HVGs. Through pre-training our model using masked expression modeling, we demonstrate its applicability to downstream tasks such as cell type classification and scRNA-seq imputation. Furthermore, we provide evidence that scHyena outperforms comparative methods in these downstream tasks across four different datasets from different brain tissues. Our contributions can be summarized as follows.

- We introduce scHyena, a model designed to handle *full-length* scRNA-seq data by leveraging the Hyena operator. To adapt the Hyena operator to our scRNA-seq analysis, we extend it into a non-causal operator, referred to as a bidirectional Hyena.

- To encode continuous scRNA-seq data, we introduce a linear layer as the adapter layer instead of discretizing and tokenizing input values. To the best of our knowledge, this is the first approach to deal with the continuous data with a Hyena operator. This approach enables us to encode scRNA-seq data without any loss of information, leading to improved performance in downstream tasks.

- In place of positional encoding in the standard Hyena, we incorporate gene encoding to provide gene-related information to the model. Although this has been demonstrated in other transformer architecture, it has not been explored before with Hyena operator and our work is the first successful demonstration.

- scHyena demonstrates superior performance compared to baseline methods in two downstream tasks. Particularly, scHyena excels in filtering out doublets in cell type classification and imputing scRNA-seq data with biologically meaningful values.

## 2 BACKGROUND

### 2.1 SELF-ATTENTION

The self-attention operator (Vaswani et al., 2017) is a fundamental mechanism of Transformers. Specifically, given a sequence $u \in \mathbb{R}^{L \times D}$ with a length of $L$, each head of the scaled self-attention operator maps $u$ to $y \in \mathbb{R}^{L \times D}$ through a self-attention operator $A(u)$:

$$A(u) = \sigma(uW_q W_k^\top u^\top), \quad y = A(u)uW_v, \tag{1}$$

where $W_q, W_k, W_v \in \mathbb{R}^{D \times D}$ represent learnable linear projections for query, key, and value, respectively, and $\sigma$ denotes the softmax and scaling operator. Through the self-attention operator, it becomes possible to capture pairwise relationships among all tokens and learn the global context of the input sequence. However, one limitation is that self-attention becomes computationally expensive for long sequences, with a complexity of $\mathcal{O}(L^2)$.

To address this computational challenge, several approaches have been developed to reduce the cost of self-attention. For instance, the factorized self-attention has been proposed for the sparse Transformer (Child et al., 2019). The factorized self-attention reduces the memory and computational requirements of self-attention by allowing self-attention heads to attend only to a subset of tokens. Another approach is the Performer (Choromanski et al., 2021) which aims to reduce the memory complexity of self-attention. By decomposing the self-attention matrix, Performer can store the implicit attention matrix with linear memory complexity, enabling it to handle longer sequences compared to Transformers with the original self-attention mechanism. However, these approaches require custom kernels that are difficult to reproduce and may involve trade-offs between memory complexity and model expressivity.

### 2.2 HYENA

A discrete convolution between an input signal $u$ with length $L$ and a convolution filter $h$ is defined as follows:

$$y_t = (h * u)_t = \sum_{\tau=0}^{L-1} h_{t-\tau} u_\tau. \tag{2}$$

Typically, in convolutional neural networks, the filter size is shorter than the input signal to manage computational complexity effectively. However, when the filter is parameterized as a function of step $t$ (i.e. $h_t = \gamma_\theta(t)$), it becomes possible to construct a long convolution filter without a large increase in the number of parameters. This type of convolution is referred to as implicit convolution.

Hyena operator (Poli et al., 2023) was introduced as a replacement for self-attention in Transformers using the implicit convolution. Specifically, the Hyena operator is characterized by a recurrent structure that involves long convolutions and element-wise gating:

$$y = x^N \cdot (h^N * (x^{N-1} \cdot (h^{N-1} * (\cdots x^1 \cdot (h^1 * v))))) \tag{3}$$

where $(v, x^1, \cdots, x^N)$ represent the projections of the input, $N$ is the number of recurrence, and $\cdot$ refers to the element-wise gating.

Fig. 2 illustrates the Hyena operator with $N = 3$. Specifically, the transformation of the input signal $u$ into the projections $(v, x^1, x^2, x^3)$ involves a linear layer and convolution operation. Subsequently, the projection $v$ undergoes long convolution with the long convolution filter $h^n$ and is subjected to element-wise gating with $x^n$. In this context, element-wise gating entails performing element-wise multiplication between the input and the corresponding projection $x^n$. Finally, the output $y$ is generated by passing the result through another linear

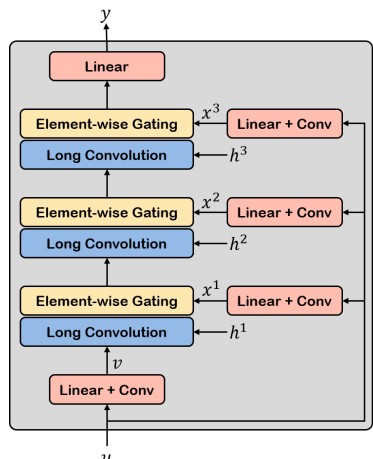

Figure 2: The Hyena operator.

layer. Notably, the convolution filters with a length $L$ are parameterized implicitly by a learnable function, enabling the performance of long convolutions without a large increase in the number

of parameters. Consequently, it becomes possible to capture long-range context without relying on self-attention. Furthermore, by performing the convolution in the Fourier domain using a fast Fourier transform (FFT), each convolution operation's time complexity is reduced to $\mathcal{O}(L \log_2 L)$. This reduced complexity allows the Hyena to handle longer sequences compared to self-attention.

## 3  scHyena

This section discusses scHyena, our novel extension of Hyena for single-cell RNA-seq analysis.

### 3.1  Extending Hyena for Full-Length RNA-seq

**Bidirectional Hyena.**  The discrete convolution shown in Eq. (1) can be also represented in matrix form as follows:

$$
y = S_h u = \begin{bmatrix} h_0 & h_{-1} & \cdots & h_{-L+1} \\ h_1 & h_0 & \cdots & h_{-L+2} \\ \vdots & \vdots & \ddots & \vdots \\ h_{L-1} & h_{L-2} & \cdots & h_0 \end{bmatrix} \begin{bmatrix} u_0 \\ u_1 \\ \vdots \\ u_{L-1} \end{bmatrix} \tag{4}
$$

where $S_h \in \mathbb{R}^{L \times L}$ is the Toeplitz matrix. In the vanilla Hyena operator, the filter $h_t$ is defined only for $t = 0, \ldots, L-1$ and has zeroes at other positions. Consequently, the output at a given position $t$ is solely dependent on the input from the past because $h_{-1}, \ldots, h_{-L+1}$ in Eq. (4) are zeroes. This inherent causality in the Hyena operator makes it suitable for applications in autoregressive language models.

However, in the context of scRNA-seq data, the notion of time causality is not relevant, and all genes can potentially have relationships regardless of their positions along the sequence. Therefore, for our model, we require a non-causal, bidirectional operator. To address this requirement, we design our convolution filters with a length of $2L - 1$, defining them for $t = -L + 1, \ldots, L - 1$. This modification allows the output $y$ to depend on the entire input across all positions.

**Linear Adaptor Layer for Expression Embedding.**  An input scRNA-seq consists of the normalized expression levels of $L$ genes, denoted as $(C_1, C_2, \ldots, C_L)$. Unlike natural language where words are tokenized into discrete tokens and each token is mapped to a unique embedding, gene expression levels are continuous values and cannot be discretized. In a previous method (Yang et al., 2022b), an attempt was made to address this issue by discretizing the expression values into bins. However, this approach carries the potential risk of information loss in the scRNA-seq data. To mitigate this concern, we encode the expression levels into expression embeddings $(E_1, E_2, \ldots, E_L)$ using a linear adapter layer, in contrast to traditional tokenization approaches. This allows us to represent gene expression levels without any loss of information.

**Gene Embedding.**  Another difference between language and scRNA-seq data is that the order of genes in scRNA-seq carries no inherent meaning. Instead, it is crucial to provide information about which gene's expression level each position in the sequence represents. To address this requirement, we incorporate gene embeddings into the scHyena model, rather than using positional encoding employed in the original Transformers. In this approach, each gene is encoded with its own embedding $(G_1, G_2, \ldots, G_L)$, which is then added to the expression embeddings. This method allows us to provide the scHyena model with explicit gene-related information.

### 3.2  Pre-Training

Fig. 3(a) illustrates the pre-training stage of scHyena model. Specifically, to pre-train our model, we employ a technique called masked expression modeling (MEM), inspired by the concept of masked language modeling (Devlin et al., 2018). Specifically, we randomly replace a subset of input embeddings with the [MASK] embedding, and then scHyena is trained to predict the expression levels of the genes that have been masked. We choose masking probability from a range of [0.05, 0.4], and we only mask nonzero values since distinguishing between true and false zero values is not feasible. As mentioned in Section 3.1, it is important to note that all genes can have relationships, irrespective of their positions. Therefore, masked expressions should be predicted by taking into consideration

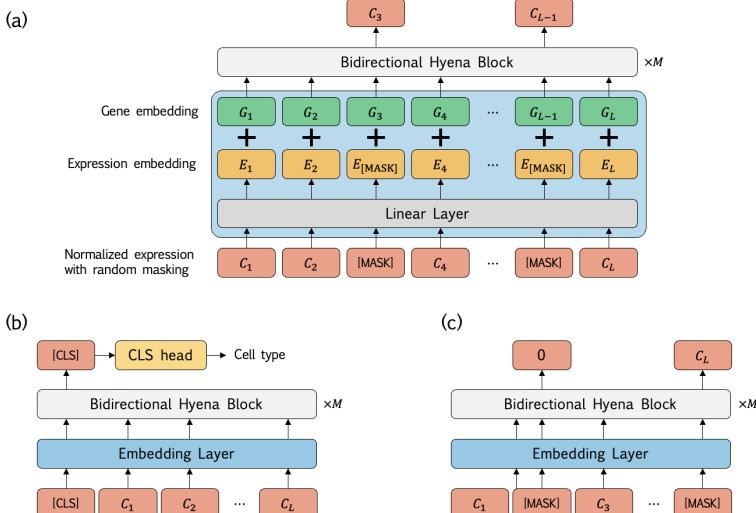

Figure 3: (a) During the pre-training stage, scHyena is trained using masked expression modeling (MEM) to acquire general representations of scRNA-seq data. The continuous expression levels are transformed into expression embeddings through a linear adaptor layer. Additionally, gene embeddings are used instead of positional encoding, which are then combined with the expression embeddings. The bidirectional Hyena blocks receive the aggregated embeddings as inputs and are trained to predict the masked expression levels. (b) During the fine-tuning process for cell type classification, we prepend the [CLS] token to the input scRNA-seq data. The embedding of the [CLS] token is then utilized as an input for the classification head, which is responsible for predicting the cell type. (c) During the fine-tuning process for scRNA-seq imputation, a relatively low masking probability is applied to zero values compared to non-zero values.

other genes, regardless of their positions. By incorporating bidirectional Hyena blocks introduced in 3.1, we empower scHyena to predict expression levels at masked positions. Specifically, the objective function for pre-training scHyena can be formulated as follows:

$$\ell_{MEM} = \sum_{i \in M} (C_i - C'_i)^2 \tag{5}$$

where $M$ represents the set of masked indices, $C_i$ and $C'_i$ denote the label and predicted gene expression level, respectively. Through this pre-training process, scHyena acquires generalizable features related to cells and genes.

## 3.3 FINE-TUNING FOR DOWNSTREAM TASKS

**Cell Type Classification.** Cell type annotation or classification is one of the most crucial tasks in scRNA-seq analysis, particularly in the context of the brain. Fig. 3(b) illustrates the fine-tuning stage of scHyena for cell type classification. To adapt the pre-trained scHyena model for cell type classification, we prepend a [CLS] token to the input scRNA-seq data. Utilizing bidirectional Hyena blocks, we enable the [CLS] token to encapsulate comprehensive information about the genes in the input. Once the input scRNA-seq data passes through the embedding layer and bidirectional Hyena blocks, the embedding of the [CLS] token is directed to a classification head. The final output of this head produces logits corresponding to cell types, and scHyena is fine-tuned using the cross-entropy loss, $\ell_{cls} = -\sum_{i=1}^{N_c} y_i \log p_i$, where $N_c$ represents the number of cell types in the data, $y_i$ denotes the cell type label, and $p_i$ signifies the Softmax probability derived from the output of scHyena.

**scRNA-Seq Imputation.** Imputing missing values in scRNA-seq data is crucial, given the prevalence of zeroes resulting from dropout events. One approach to imputation is to directly adapt the pre-training strategy. However, in pre-training, zero values are not masked, potentially causing the model to learn to replace true zero values with other values. Alternatively, if we randomly mask zero values with the same probability as non-zero values, the model may lean towards outputting

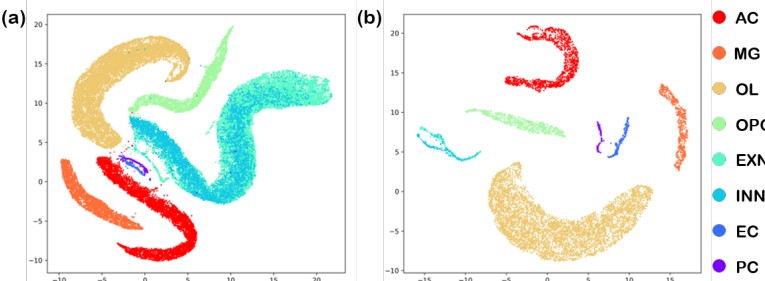

Figure 4: The UMAP visualization results of cell embeddings obtained from the pre-trained scHyena model on two datasets (AC: astrocyte, MG: microglia, OL: oligodendrocyte, OPC: oligodendrocyte progenitor cell, EXN: excitatory neuron, INN: inhibitory neuron, EC: endothelial cell, PC: pericyte). (a) *Lau* dataset. (b) *Smajic* dataset.

zeroes predominantly. This is due to the fact that the majority of values in scRNA-seq data are zeroes. To address this issue, we differentiate the masking probabilities for zero values and non-zero values. Specifically, we apply a masking probability of 0.4 to non-zero values, while zero values are masked with a probability of 0.04. This helps to balance the masking of zero and non-zero values. However, a challenge arises in distinguishing true zero values from false zeroes (dropout), and the model may impute false zeroes as actual zeroes. Fortunately, our scHyena model incorporates gene embeddings, which enables it to learn the expression level tendencies of each gene. Fig. 3(c) illustrates the fine-tuning stage for scRNA-seq imputation, and the loss function for the imputation task is the same as the pre-training loss function, Eq. (5).

## 4 EXPERIMENTS

**Dataset.**  To pre-train our model, we utilize two publicly available brain scRNA-seq datasets from previous studies (Kamath et al., 2022; Wang et al., 2022). For our downstream tasks, we evaluate the performance of our method on four additional datasets, referred to as the *Lau* (Lau et al., 2020), *Leng* (Leng et al., 2021), *Smajic* (Smajić et al., 2022), and *Zhu* (Zhu et al., 2022) datasets. To ensure consistency across all datasets, we process them uniformly by mapping gene IDs to Ensembl stable gene IDs, resulting in datasets containing expression information for 19,306 genes with unique Ensemble IDs. As part of our preprocessing pipeline for scRNA-seq data, we initially filtered out cells with a total gene expression level of less than 200. Subsequently, we normalize the gene expression values so that the total count of gene expressions of each cell is set to 10,000. Finally, we apply log normalization ($\log(x + 1)$) to obtain the final pre-process data. For more detailed information about our datasets, please refer to Appendix A.

**Pre-Training.**  We implemented our model based on the official source code of Hyena and HyenaDNA (Poli et al., 2023; Nguyen et al., 2023). To construct scHyena model, we set $N$ as 3 in Eq. (3) and stack four bidirectional Hyena blocks ($M = 4$ in Fig. 3). For the pre-training phase, we trained the model for 2 epochs using AdamW optimizer (Loshchilov & Hutter, 2019) with a learning rate of 1e-4. The pre-training took approximately 3.5 days with a batch size of 8 in two RTX 3090 units. To demonstrate that the scHyena has learned meaningful cell representations during pre-training, we extract the cell features from the pre-trained scHyena model and visualize them in a 2D space using UMAP (McInnes et al., 2018). Specifically, we obtain the model's output with the shape of $L \times D$, where $L$ represents the length of scRNA-seq data, and $D$ is the embedding dimension of the model. We then compute the average along the first dimension, resulting in cell embeddings of dimension $D$ for each cell.

Fig. 4 displays the UMAP embedding results on two datasets that were not utilized during pre-training. As depicted in Fig. 4(a) (*Lau* dataset), most cell embeddings form clusters with other embeddings representing the same cell type, except for excitatory neurons (EXN) and inhibitory neurons (INN). Notably, this occurs even though no cell type information was provided to the scHyena model during the pre-training phase. In the case of the excitatory and inhibitory neurons, while they are not distinctly separated on the UMAP plot, it is evident that there are regions within the cluster where each neuron type exhibits greater cohesion. Furthermore, Fig. 4(b) demonstrates more clearly separated results when using the *Smajic* dataset. These findings provide convincing evidence

| Experiment | Data | Method | F1-score | | | | | | | | | Macro | Micro | Weighted |
|---|---|---|---|---|---|---|---|---|---|---|---|---|---|---|
| | | | Cell Type | | | | | | | | | | | |
| | | | AC | MG | OL | OPC | EXN | INN | EC | PC | DT | | | |
| With DT | Lau | Seurat | 0.909 | 0.922 | 0.959 | 0.860 | 0.931 | 0.939 | 0 | 0.594 | 0.435 | 0.875 | 0.728 | 0.872 |
| | | SciBet | 0.968 | 0.974 | 0.981 | 0.970 | 0.988 | 0.983 | 0.650 | **0.949** | 0.816 | 0.964 | 0.920 | 0.962 |
| | | scBERT | 0.984 | 0.981 | **0.990** | **0.987** | 0.990 | 0.984 | **0.942** | 0.913 | 0.911 | 0.979 | **0.965** | 0.979 |
| | | scHyena | **0.988** | **0.991** | **0.990** | 0.981 | **0.992** | **0.985** | 0.916 | 0.917 | **0.923** | **0.982** | **0.965** | **0.982** |
| | Leng | Seurat | 0.977 | 0.972 | 0.981 | 0.981 | 0.965 | 0.985 | 0 | 0.617 | 0.382 | 0.950 | 0.762 | 0.949 |
| | | SciBet | 0.978 | 0.967 | 0.988 | 0.972 | 0.987 | **0.990** | 0.836 | **0.952** | 0.495 | 0.972 | 0.907 | 0.967 |
| | | scBERT | 0 | 0 | 0 | 0 | 0.558 | 0 | 0 | 0 | 0 | 0.387 | 0.062 | 0.216 |
| | | scHyena | **0.991** | **0.985** | **0.992** | **0.986** | **0.992** | 0.989 | **0.857** | **0.952** | **0.789** | **0.984** | **0.948** | **0.983** |
| | Smajic | Seurat | 0.982 | 0.971 | 0.984 | 0.990 | 0.629 | 0.756 | 0.973 | 0.770 | 0.300 | 0.943 | 0.817 | 0.944 |
| | | SciBet | 0.987 | 0.995 | 0.993 | 0.976 | 0.780 | 0.755 | 0.973 | 0.783 | 0.293 | 0.962 | **0.937** | 0.953 |
| | | scBERT | 0 | 0 | 0.758 | 0 | 0 | 0 | 0 | 0 | 0 | 0.610 | 0.084 | 0.463 |
| | | scHyena | **0.997** | **0.998** | **0.997** | **0.993** | **0.840** | **0.821** | **0.982** | **0.958** | **0.844** | **0.984** | **0.937** | **0.983** |
| | Zhu | Seurat | 0.964 | 0.964 | 0.988 | 0.810 | 0.986 | 0.981 | 0.953 | 0.976 | 0.248 | 0.945 | 0.874 | 0.943 |
| | | SciBet | 0.984 | 0.990 | 0.994 | 0.988 | 0.990 | 0.988 | 0.960 | 1 | 0.721 | 0.982 | 0.957 | 0.980 |
| | | scBERT | 0 | 0 | 0.378 | 0 | 0 | 0 | 0 | 0 | 0 | 0.233 | 0.042 | 0.088 |
| | | scHyena | **0.991** | **0.992** | **0.996** | **0.995** | **0.992** | **0.991** | **0.979** | 1 | **0.833** | **0.988** | **0.974** | **0.987** |
| Without DT | Lau | Seurat | 0.999 | 0.999 | 1 | **0.999** | 0.997 | 0.993 | 0.992 | 0.99 | - | **0.998** | **0.996** | **0.998** |
| | | SciBet | 1 | 1 | 1 | **0.999** | **0.998** | 0.992 | 0.975 | 0.983 | - | **0.998** | 0.993 | **0.998** |
| | | scBERT | 0 | 0 | 0 | 0 | 0.543 | 0 | 0 | 0 | - | 0.373 | 0.068 | 0.202 |
| | | scHyena | 0.999 | 0.999 | 1 | 0.998 | **0.998** | **0.994** | 0.967 | 0.979 | - | **0.998** | 0.992 | **0.998** |
| | Leng | Seurat | 0.999 | **0.999** | 0.999 | **0.996** | 0.996 | 0.992 | 1 | 0.967 | - | 0.996 | 0.993 | 0.996 |
| | | SciBet | 1 | **0.999** | 0.999 | 0.995 | **0.998** | **0.995** | 1 | 0.984 | - | **0.998** | **0.996** | **0.998** |
| | | scBERT | 0 | 0 | 0 | 0 | 0.573 | 0 | 0 | 0 | - | 0.402 | 0.073 | 0.230 |
| | | scHyena | 0.999 | 0.997 | 1 | 0.990 | **0.998** | 0.994 | 0.978 | 0.967 | - | 0.997 | 0.990 | **0.997** |
| | Smajic | Seurat | 1 | 1 | 1 | 1 | 0.850 | 0.756 | 1 | 1 | - | 0.992 | 0.951 | 0.992 |
| | | SciBet | 1 | 1 | 1 | 1 | 0.852 | 0.758 | 1 | 1 | - | 0.992 | 0.951 | 0.992 |
| | | scBERT | 0 | 0 | 0.775 | 0 | 0 | 0 | 0 | 0 | - | 0.633 | 0.097 | 0.491 |
| | | scHyena | 1 | 1 | 1 | 1 | **0.891** | **0.851** | 0.997 | 0.995 | - | **0.994** | **0.967** | **0.994** |
| | Zhu | Seurat | 1 | **0.999** | 0.999 | 0.999 | 0.997 | **0.995** | 1 | 0.988 | - | **0.998** | 0.997 | 0.998 |
| | | SciBet | 1 | 0.998 | 1 | 1 | 0.996 | 0.991 | 1 | 1 | - | 0.997 | **0.998** | 0.997 |
| | | scBERT | 0 | 0 | 0.389 | 0 | 0 | 0 | 0 | 0 | - | 0.242 | 0.049 | 0.094 |
| | | scHyena | 0.999 | 0.998 | 1 | 0.999 | **0.998** | **0.995** | 0.993 | 0.988 | - | **0.998** | 0.996 | **0.998** |

Table 1: Cell type classification results on various methods (AC: astrocyte, MG: microglia, OL: oligodendrocyte, OPC: oligodendrocyte progenitor cell, EXN: excitatory neuron, INN: inhibitory neuron, EC: endothelial cell, PC: pericyte, DT: doublet).

that our scHyena model learns meaningful cell representations during the pre-training phase. The UMAP plots for the other two datasets can be found in the Appendix D.

**Cell Type Classification.** scRNA-seq techniques often process cells individually, but sometimes multiple cells are captured in the same reaction, forming hybrid transcriptomes called doublets (Macosko et al., 2015; Zheng et al., 2017; Cao et al., 2017). Identifying doublets typically relies on unique molecular identifier (UMI) counts or the presence of multiple marker genes. However, accurate detection can be challenging due to cellular diversity and overlap with intermediate cell states expressing markers of multiple types. In the scHyena model, we have also incorporated a doublet detection mode as a part of cell type classification.

To assess the performance of scHyena in the cell type classification task, we conducted a comparative analysis with three baseline methods: Seurat (Hao et al., 2021), SciBet (Li et al., 2020), and scBERT (Yang et al., 2022b). To ensure the reproducibility of our results on our datasets, we referred to the official source codes of these baseline methods. Furthermore, in experiments involving doublets, we employed DoubletFinder (McGinnis et al., 2019) in conjunction with Seurat for doublet identification.

Table 4 displays the F1-scores for each cell type, as well as the macro, micro, and weighted averages of the F1-scores for cell type classification methods. Overall, scBERT exhibits poor performance, primarily due to its tendency to output only one class in most cases. It appears that scBERT is not robust for imbalanced datasets, as it frequently predicts the most common cell types in the dataset (the distribution of each cell type in the datasets is provided in Table A in Appendix A). In experiments involving doublets, Seurat struggles to classify endothelial cells (EC) within the *Lau* and *Leng* datasets. This issue arises because DoubletFinder missed some doublet samples before the clustering process in Seurat. As a result, the presence of remaining doublets complicates the separation of the clusters of endothelial cells and pericytes (PC), both of which are vascular cells. In comparison to Seurat, SciBet exhibits relatively better performance in filtering out doublets from the *Lau* and *Zhu* datasets and performs well in other cell types. However, in the case of the *Leng* and *Smajic* datasets, SciBet struggles to filter out doublets, leading to an overall drop in performance. On the other hand, scHyena demonstrates the highest F1-scores in most cases in experiments involving doublets. It excels at filtering out doublets with high accuracy, resulting in an overall enhancement of F1-scores in cell type classification. Furthermore, even when doublets are excluded from the ex-

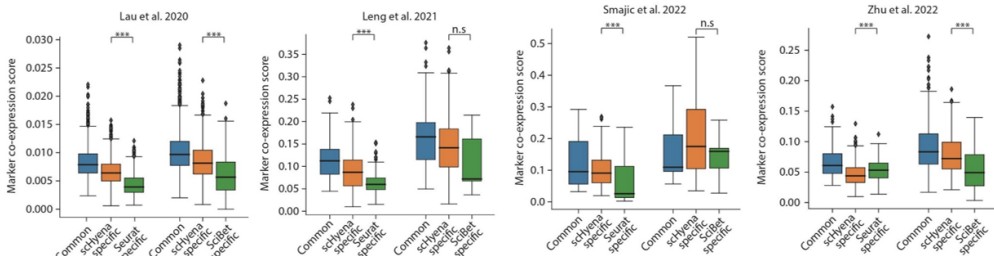

Figure 5: Box plot for co-expression scores between 40 cell type marker genes along the group of doublets. For statistical significance test, two-sided Welch's t-test was performed for each pair of type of doublet, corrected by Benjamini Hochberg (∗: P < 0.05, ∗∗: P < 0.01, ∗ ∗ ∗: P < 0.001, n.s: non-significant).

periments, scHyena still delivers similar or superior performance compared to the baseline methods. Confusion matrices for the cell type classification experiments are available in Appendix D.

Given the absence of the ground-truth reference, we further investigated about whether scHyena detect better doublet-like feature compared to other tools in more detail. To estimate the doublet-like feature without the reference, we adopt a marker gene co-expression scoring scheme from Scrublet (Wolock et al., 2019), which measures the associated pattern of expression level between each pair of genes. We compare scHyena to other tools by dividing doublets into three annotation groups - common, scHyena specific, and other algorithm specific - and measure the co-expression score levels of cell type specific marker genes in each group. In Fig. 5, commonly specific doublet groups show the highest scores in most cases of comparison. Moreover, the scHyena specific group presented higher co-expression scores compared to other tool specific groups. Specifically, scHyena significantly shows higher scores than Seurat in all data except *Zhu*. The median co-expression scores were also higher than the score of SciBet in all four datasets, although two datasets, *Leng* and *Smajic* show statistically non-significant due to the small size of the SciBet specific doublet group. In total, scHyena could distinguish out doublet-like features more robustly than other doublet classification tools.

**scRNA-Seq Imputation.** To assess the imputation performance of scHyena, we conducted comparisons with baseline methods, MAGIC (Van Dijk et al., 2018) and DCA (Eraslan et al., 2019). For the quantitative evaluation, we employed the following procedure: we masked non-zero values in the input data and applied the imputation methods to predict these masked values. To ensure a comprehensive evaluation, we divided the non-zero indices into five sub-groups and assessed the imputation methods independently for each sub-group. This assessment involved measuring the Mean Squared Error (MSE) and Pearson correlation coefficient between true values and imputed values at masked indices.

Fig. 6 presents joint plots comparing true values and imputed values along with MSEs and Pearson correlation coefficients for each group in the *Smajic* dataset. The figure clearly demonstrates that scHyena outperforms both the MAGIC and DCA methods in terms of MSE and Pearson correlation coefficients. Notably, the joint plots reveal that scHyena exhibits a distribution similar to the true values, unlike other baseline methods. In the joint plots of scHyena, most dots align near the $y = x$ graph, while the dots in the joint plots of other methods fall below the $y = x$ line. These findings strongly indicate that scHyena excels in imputing non-zero values with significantly lower error compared to other methods. Joint plots for other datasets can be found in Appendix D.3.

For a more in-depth analysis of the imputation performance of scHyena, we impute zero values in the scRNA-seq data using various methods and then project them into a 2D space using UMAP. In theory, if the zero values are imputed with appropriate values, the samples should form denser clusters with other samples of the same cell type when the imputed scRNA-seq data is projected.

Fig. 7 illustrates the UMAP plots of raw and imputed scRNA-seq data for the *Smajic* dataset. In the first column, even though cells of the same types are clustered together, cells from different batches (patients) are far apart when projecting the raw counts. Notably, oligodendrocytes (OL) exhibit noticeable batch effects in Fig. 7(b), indicating that the cells are grouped more by technical arrangement than biological factors. Conversely, MAGIC corrects the batch effect and helps group cells of the same types by imputing zero values. However, in the UMAP of imputed samples by

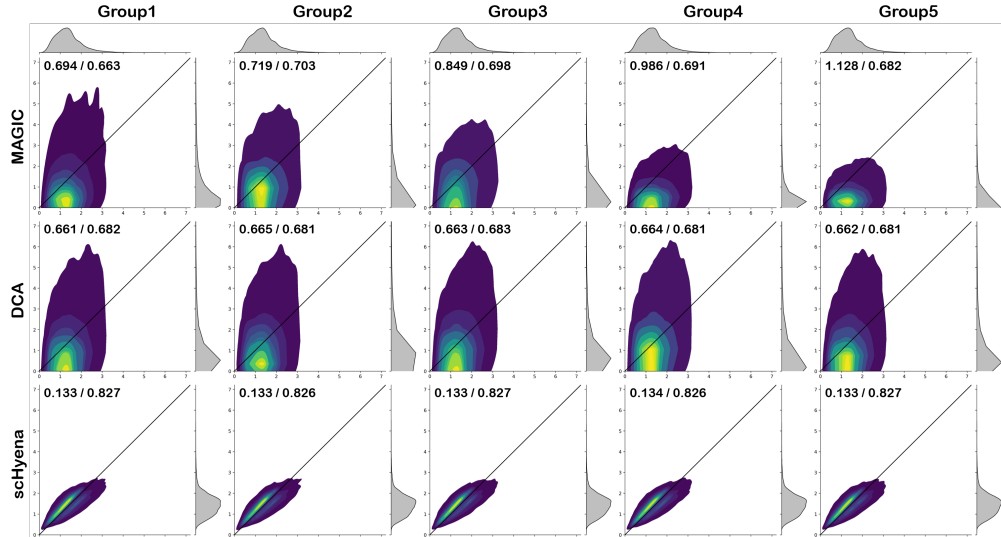

Figure 6: Joint plots comparing true values and imputed values predicted by the imputation methods on the *Smajic* dataset (x-axis: true values, y-axis: imputed values). The values in the upper left corner of each scatter plot represent MSEs and Pearson correlation coefficients.

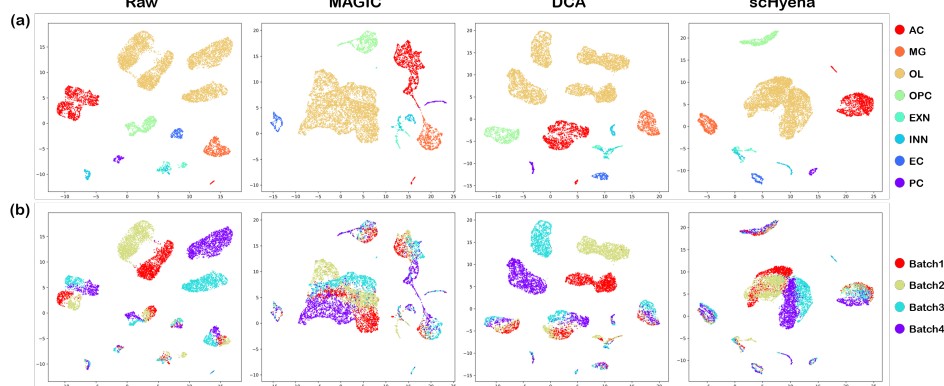

Figure 7: The UMAP visualization of raw and imputed scRNA-seq data for the *Smajic* dataset. The figures are labeled with (a) the cell type and (b) batch (patient).

MAGIC, the clusters of astrocytes (AC) and microglia (MG) are not completely separated in some regions. Regarding DCA, the batch effects in oligodendrocytes remain uncorrected, suggesting that the imputed values by DCA are not accurate. In contrast, when scRNA-seq data imputed by scHyena are projected, they form dense clusters with cells of the same brain cell type. Particularly, as shown in the last column of Fig. 7(a) and (b), the oligodendrocytes form the densest cluster compared to other columns, even though they come from different batches. This indicates that scHyena imputes zero values with biologically meaningful counts, leading to the correction of batch effects. For more UMAP visualization results, please refer to Appendix D.3.

## 5 CONCLUSION

In this study, we introduced scHyena, a foundation model for scRNA-seq analysis of brain tissue. We leveraged our pre-trained scHyena model for key downstream tasks: cell type classification, doublet detection, and scRNA-seq imputation. Our extensive experiments demonstrated that our proposed method consistently outperforms existing baseline methods in both cell type classification and scRNA-seq imputation. While we specifically applied scHyena to a few downstream tasks, its utility may extend to a broader spectrum of brain-related applications, including providing valuable insights into diseases such as Alzheimer's or Parkinson's, which is our future research scope. Additionally, by pre-training scHyena with scRNA-seq data from various tissue types, we anticipate its applicability expanding to a wider array of downstream tasks.

ETHICS STATEMENT

The utilization of foundation models such as scHyena offers significant benefits for advancing our understanding of complex biological systems and potentially improving medical research. However, ethical considerations must guide its use to ensure responsible data handling, avoid biases, and protect individual privacy, underscoring the importance of ethical guidelines and regulations in the application of such models.

REPRODUCIBILITY STATEMENT

We provide detailed implementation information in Section 4 and additional details in Appendix B. A comprehensive description of the datasets used in our experiments can be found in Section 4 and Appendix A. Our source code is available for access at the following link: https://github.com/scHyena2023/scHyena.

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

# A  DATASET

**Collection of Published Human Brain scRNA-Seq Data.** For published data, 14 distinct snRNA-seq processed data were collected in the form of gene by cell count matrices, with the following identifiers; GSE140231 (Agarwal et al., 2020), GSE148822 (Gerrits et al., 2021), GSE178265 (Kamath et al., 2022), GSE157827 (Lau et al., 2020), GSE147528 (Leng et al., 2021), GSE129308 (Otero-Garcia et al., 2022), GSE174367 (Morabito et al., 2021), GSE167494 (Sadick et al., 2022), GSE157783 (Smajić et al., 2022), GSE160936 (Smith et al., 2022), GSE184950 (Wang et al., 2022), GSE163577 (Yang et al., 2022a), GSE188545 (Zhang et al., 2023), GSE202210 (Zhu et al., 2022).

To integrate the data along the same transcriptome information, Ensembl stable gene id was used instead of the gene symbol. We concatenated all the data along the union set of Ensembl id, where 61,325 genes remained. The undetected genes in each cell were set as 0. Genes of chromosome Y or without annotation information in GRCh38 version 108 GTF file were first filtered out. Genes detected at least one for more than 0.5% of whole cells except for *Kamath* dataset were only selected as target genes, remaining 19,306 unique Ensembl id.

Among the data, six different types of data were selected as datasets for scHyena model; *Kamath* (Kamath et al., 2022), *Wang* (Wang et al., 2022) for model pre-training, *Lau* (Lau et al., 2020), *Leng* (Leng et al., 2021), *Smajic* (Smajić et al., 2022), *Zhu* (Zhu et al., 2022) for downstream tasks. These data were selected based on the following considerations. *Kamath* and *Wang* datasets, which contain the largest size of nuclei, were selected for model pre-training. Since both data are enriched with specific diseases and tissues (Parkinson's disease and substantia nigra), we design test data to cover diverse types of tissues and diseases, especially for the same number of Alzheimer's disease and Parkinson's disease to compensate for the biases. *Lau* and *Leng* datasets were selected for the case of Alzheimer's disease, covering three different types of the tissue-prefrontal cortex, caudal entorhinal cortex, and superior frontal gyrus while preserving the large size of nucleus. For Parkinson's disease, *Smajic* and *Zhu* datasets were selected, covering the substantia nigra and frontal cortex, respectively.

| Dataset | | AC | MG | OL | OPC | EXN | INN | EC | PC | DT | ETC | Total (train/test) |
|---|---|---|---|---|---|---|---|---|---|---|---|---|
| Pre-training | *Kamath* | 40,848 | 34,816 | 185,451 | 15,148 | 28,031 | 11,570 | 5,786 | 3,927 | 5,603 | 99,132 | 430,312 (430,312/0) |
| | *Wang* | 5,942 | 7,803 | 81,378 | 8,361 | 5,987 | 840 | 4,804 | 2,137 | 21,202 | 6,716 | 145,170 (145,170/0) |
| Downstream Tasks | *Lau* | 12,157 | 4,719 | 30,571 | 9,223 | 50,572 | 18,932 | 540 | 444 | 12,427 | - | 139,585 (99,801/39784) |
| | *Leng* | 6,650 | 2,260 | 11,904 | 3,038 | 19,926 | 9,656 | 194 | 135 | 3,590 | - | 57,353 (47,269/10,084) |
| | *Smajic* | 5,018 | 3,717 | 20,956 | 2,674 | 980 | 705 | 194 | 1,641 | 1,479 | - | 37,364 (29,491/7,873) |
| | *Zhu* | 7,077 | 4,386 | 22,773 | 4,737 | 19,200 | 11,763 | 233 | 156 | 3,425 | - | 73,750 (57,729/16,021) |

Table 2: The distribution of cell types in the datasets (AC: astrocyte, MG: microglia, OL: oligodendrocyte, OPC: oligodendrocyte progenitor cell, EXN: excitatory neuron, INN: inhibitory neuron, EC: endothelial cell, PC: pericyte, DT: doublet, ETC: others).

**Cell Type Annotation Based on Unsupervised Clustering.** The collected count matrices were preprocessed based on the canonical SCANPY analysis pipeline (Wolf et al., 2018). Produced data were integrated with the collected public data along the 19,306 unified list of genes. Patients with less than 200 cells were also filtered out, a total of 2,408,023 nuclei from 461 patients remained. To distinguish the doublet produced by the experimental error of the single-cell technique, we perform Scrublet (Wolock et al., 2019) to calculate the doublet score and predict the doublet of each single cell. Doublet score was used for annotating doublet-enriched clusters. To cluster cells into each cell type, the top 2,000 highly variable genes were first selected based on analytic Pearson residuals (Lause et al., 2021) and used for the computation of PCA coordinates and clustering. The total UMI count sum was normalized to be equal to 50,000 for each single cell, log2-transformed with pseudo-count 1. Normalized data of highly variable genes were then scaled by scanpy.pp.scale and PCA coordinates were computed by scanpy.pp.pca with default parameters. To remove the confounded factors between patients, Harmony correction (Korsunsky et al., 2019) was performed on PCA coordinates across patients of single cells. Neighborhoods of each single cell were calculated by scanpy.pp.neighbors with the parameter of 20 PC components and 40 nearest neighbors. The data were then reduced on the 2-dimensional plane through UMAP. Leiden clustering was performed on resolution 1.8 to finally distinguish 69 distinct clusters.

To annotate the cell type for each cluster, the expression level of known marker genes for major cell types in the human brain was investigated along those clusters. By checking 40 distinct marker genes, 51 clusters were annotated to 11 distinct cell types including 8 major brain cell types; Oligo-

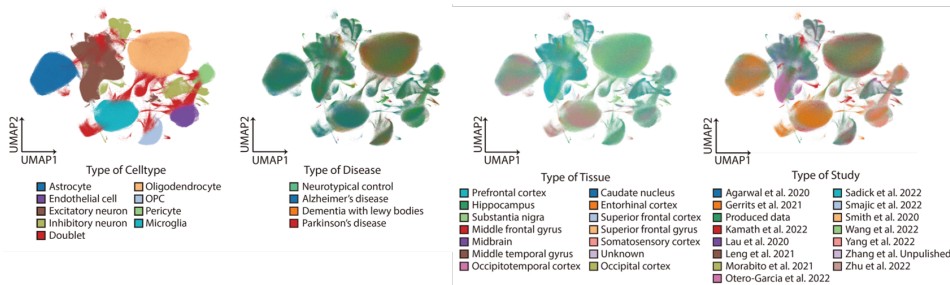

Figure 8: Distribution of four different types of meta data on UMAP for 8 major cell types.

dendrocyte (CLDN11 and MBP), Astrocyte (AQP4 and ALDH1L1), Microglia (C1QC and CSF1R), Endothelial cell (CLDN5 and FLT1), Pericyte (PDGFRB), Oligodendrocyte progenitor cell (OPC; PDGFRA and VCAN), Excitatory neuron (SYT1, SLC17A7, and SLC17A6) and Inhibitory neuron (SYT1, GAD1, and GAD2), and 3 subtypes; Neuron subtype 1 (SYT1, SLC17A6, and GAD2), Neuron subtype 2 (SYT1, neither SLC17A6 nor GAD2), and Myeloid subtype 1 (GNLY and CD44). One cluster that did not show a specific expression pattern toward a list of marker genes was annotated as unidentified. 17 Clusters that showed an average of doublet score more than 0.1 were annotated as Doublet. Among the clusters, a single cell annotated as one of 8 major cell types and doublets were used for the cell type classification task of the model (Fig. 8). There were no clustering biases on non-neuronal cells, presenting homogeneous clusters. Excitatory neurons were forms of several distinguishable clusters, which may be due to the diversity of neuron populations compared to glial cells.

## B  EXPERIMENTAL DETAILS

### B.1  DOWNSTREAM TASKS

For all downstream tasks, we fine-tuned the model over 5 epochs using the AdamW optimizer (Loshchilov & Hutter, 2019) with a learning rate of 1e-5. The fine-tuning process was executed using two RTX 3090 units. Additionally, we set aside ten percent of the training set as a validation set. The model with the best performance on the validation set was selected for inference.

**Cell Type Classification.**  For cell type classification, we employ the embedding of the [CLS] token as the input for the classification head, which consists of a linear layer. Since scRNA-seq data is not composed of discrete tokens, directly prepending the [CLS] token to the input scRNA-seq data is not feasible. Instead, as demonstrated in the Vision Transformer (Dosovitskiy et al., 2020), we prepend a learnable [CLS] embedding to the embeddings of the input scRNA-seq data.

We rely on samples where the cell type has been identified in the cell type classification task to ensure accurate performance evaluation. We partition each dataset for downstream tasks into training and testing sets, with the number of cells in each set detailed in Table A. In experiments excluding doublets, we exclude all cells labeled as doublets, omitting them from both the training and testing phases.

**scRNA-Seq Imputation.**  As previously mentioned in the main text, we conducted two experiments to evaluate the imputation performance of scHyena. In the first experiment, we initially divided non-zero values into five groups. Subsequently, we masked each group separately and conducted imputation for quantitative evaluation. To assess the imputation performance, we calculated the Mean Squared Error (MSE) and Pearson correlation coefficient between the true values and the imputed values. All MSEs and Pearson correlation coefficients were computed based on normalized counts after log normalization. Subsequently, we imputed zero values in the scRNA-seq dataset and assessed the impact on cell clustering. Similar to the first experiment, we divided the zero values into ten sub-groups and performed separate imputations for each sub-group. After each imputation, the imputed sequences were merged to create fully imputed scRNA-seq data.

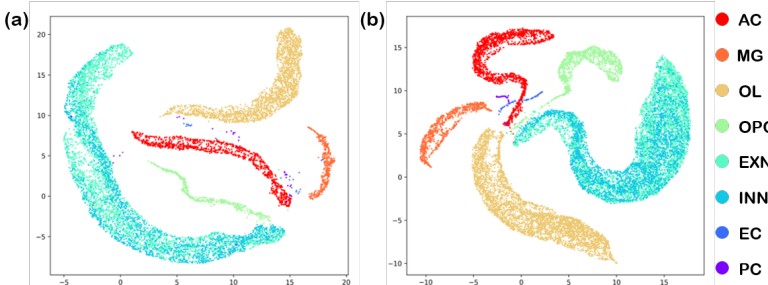

Figure 9: The UMAP visualization results of cell embeddings obtained from the pre-trained scHyena model on two datasets (AC: astrocyte, MG: microglia, OL: oligodendrocyte, OPC: oligodendrocyte progenitor cell, EXN: excitatory neuron, INN: inhibitory neuron, EC: endothelial cell, PC: pericyte). (a) *Leng* dataset. (b) *Zhu* dataset.

**UMAP.** Uniform Manifold Approximation and Projection for Dimension Reduction (UMAP) (McInnes et al., 2018) is a widely-used technique for reducing the dimensionality of high-dimensional data and visualizing it in a lower-dimensional space. In our experiments, we employed UMAP. For UMAP, we configured the hyperparameters with $n\_neighbors = 15$ and $min\_dist = 0.5$.

## C BASELINE METHODS

### C.1 CELL TYPE CLASSIFICATION

**Seurat.** Seurat (Hao et al., 2021) is one of the popular tools for single-cell analysis. In our analysis, we followed the official tutorial for cell clustering. After clustering, we manually assigned cell types to each cluster by comparing the marker genes of each cell type with the marker genes of each cluster. In experiments involving doublets, we utilized DoubletFinder (McGinnis et al., 2019) in conjunction with Seurat. Both Seurat and DoubletFinder are implemented in R, and we used Seurat version 4.3.0.

**SciBet.** SciBet (Li et al., 2020) is a supervised cell type annotation method designed for scRNA-seq data. To classify cell types using SciBet, we followed the tutorial provided by the authors. For training SciBet, we utilized the same training set as used for training our scHyena model.

**scBERT.** scBERT (Yang et al., 2022b) is a pre-trained model for cell type annotation of scRNA-seq data based on the Performer (Choromanski et al., 2021). Similar to scHyena, scBERT also employs a pre-training strategy that involves masking some nonzero expression levels. Additionally, scBERT also uses the entire expression level data without reducing the number of genes because it is based on the Performer, which provides reduced memory complexity. However, a key difference is that scBERT discretizes the expression levels by binning them, whereas we use the expression levels without any discretization. Furthermore, because the pre-trained weights provided by the authors of scBERT were not specifically trained for brain cells and the list of genes differed from our dataset, we conducted our own pre-training of the scBERT model using the same datasets that were used for pre-training scHyena. The official code for scBERT only provides code for fine-tuning, so we re-implemented the pre-training code based on the official fine-tuning code and used the official code for fine-tuning the model for cell type classification.

### C.2 SCRNA-SEQ IMPUTATION

**MAGIC.** Markov Affinity-based Graph Imputation of Cells (MAGIC) (Van Dijk et al., 2018) is an algorithm designed for denoising scRNA-seq data. In our implementation, we utilized the Python version of MAGIC and followed the guidelines outlined in the official tutorial.

Figure 10: Dot plots displaying the expression patterns of marker genes for 8 major types of brain cells in cells classified as pericytes by each method. The expression values were log2 transformed, scaled for each study, and then averaged.

**DCA.** Deep Count Autoencoder (DCA) (Eraslan et al., 2019) is the method designed for denoising scRNA-seq data through the use of a zero-inflated negative binomial loss function. To implement DCA, we utilized the official source code and followed the tutorial provided by the authors.

## D   ADDITIONAL RESULTS

### D.1   CELL EMBEDDING OF PRE-TRAINED SCHYENA

In addition to the results in Section 3.3, Fig. 9 presents the UMAP visualization of cell embeddings generated by pre-trained scHyena on two additional datasets. As observed in Fig. 4, the majority of cells cluster together with cells of the same type, except for excitatory and inhibitory neurons, which form a joint cluster.

### D.2   CELL TYPE CLASSIFICATION

Fig. 10 displays the expression patterns of marker genes in cells that classified to pericytes, categorized into common, scHyena specific, and other specific groups. As depicted in Fig. 10, the scHyena specific groups exhibit relatively high expression level in pericyte marker genes, while endothelial marker genes are highly expressed in the Seurat specific or SciBet specific groups.

Figs. 11 and 12 show the confusion matrices of cell type classification methods on various datasets. Again, scHyena shows outstanding performance on cell type classification.

### D.3   SCRNA-SEQ IMPUTATION

Fig. 13 depicts the joint plots comparing true values with values imputed by various imputation methods. Figs. 14 to 16 display the UMAP plots of raw and imputed scRNA-seq data for the *Lau*, *Leng*, and *Zhu* datasets, respectively.

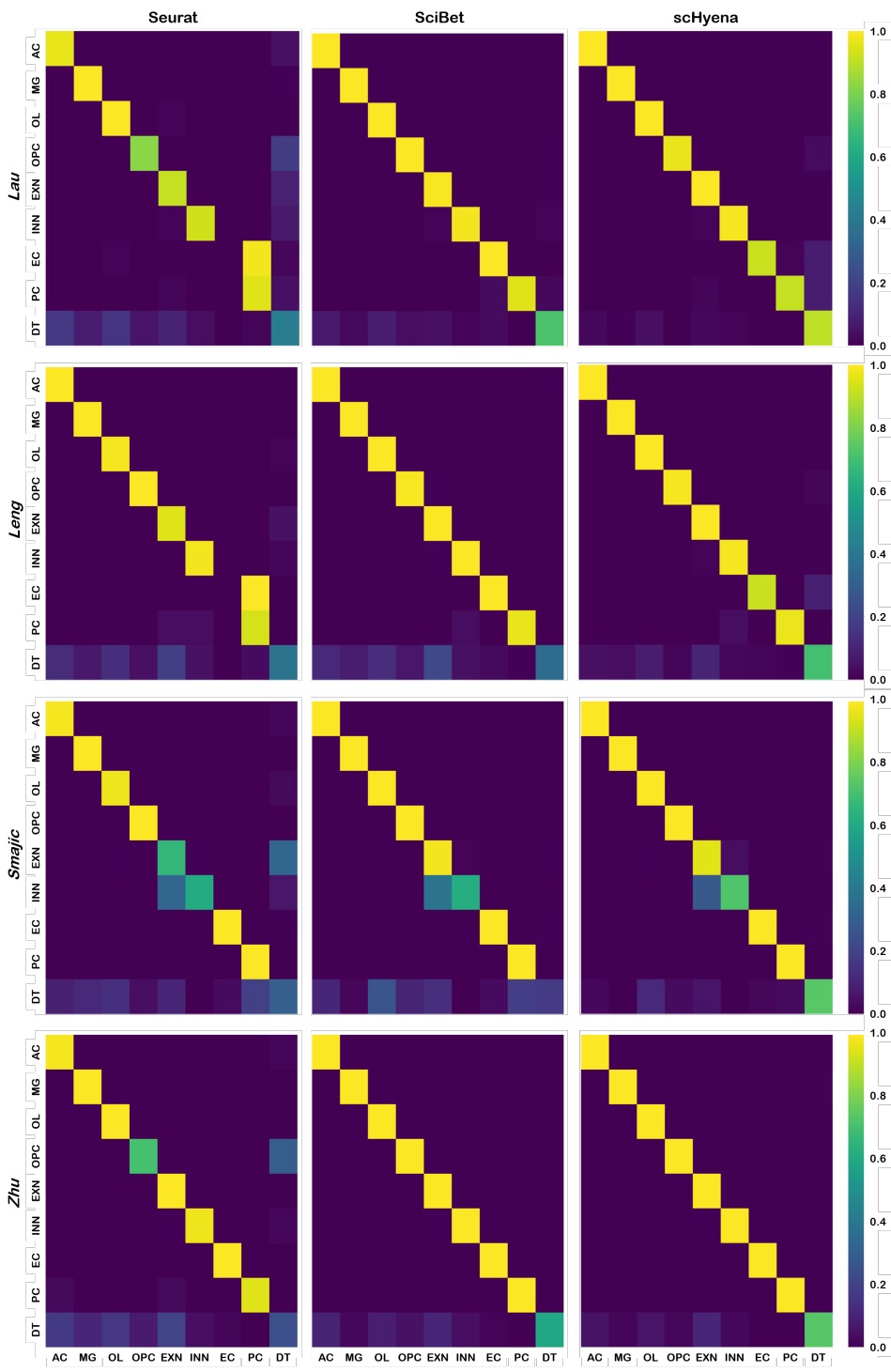

Figure 11: Confusion matrices of cell type classification methods on the *Lau*, *Smajic*, and *Zhu* datasets with doublets (rows: labels, columns: predictions).

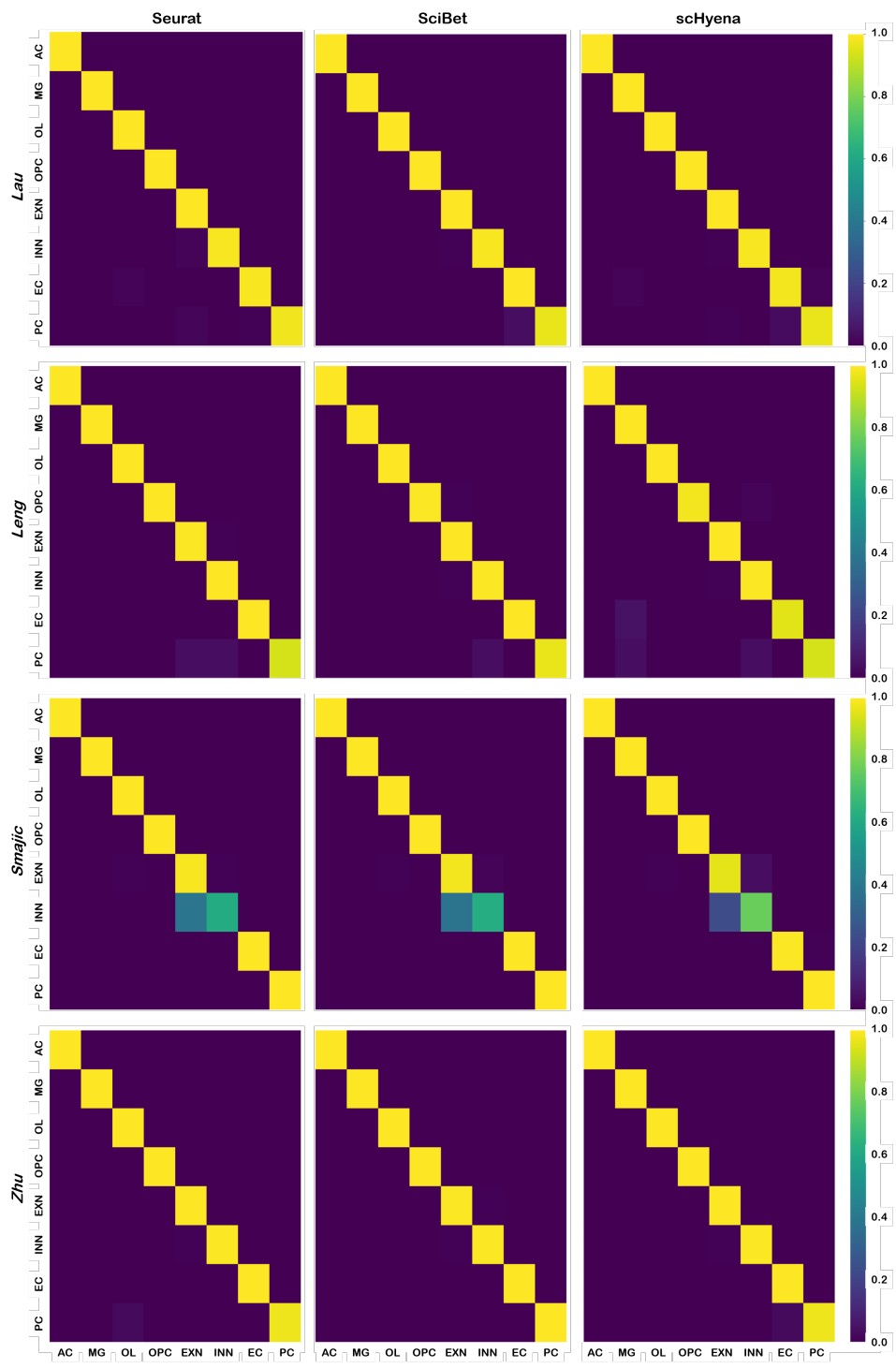

Figure 12: Confusion matrices of cell type classification methods on the *Lau*, *Leng*, *Smajic*, and *Zhu* datasets without doublets (rows: labels, columns: predictions).

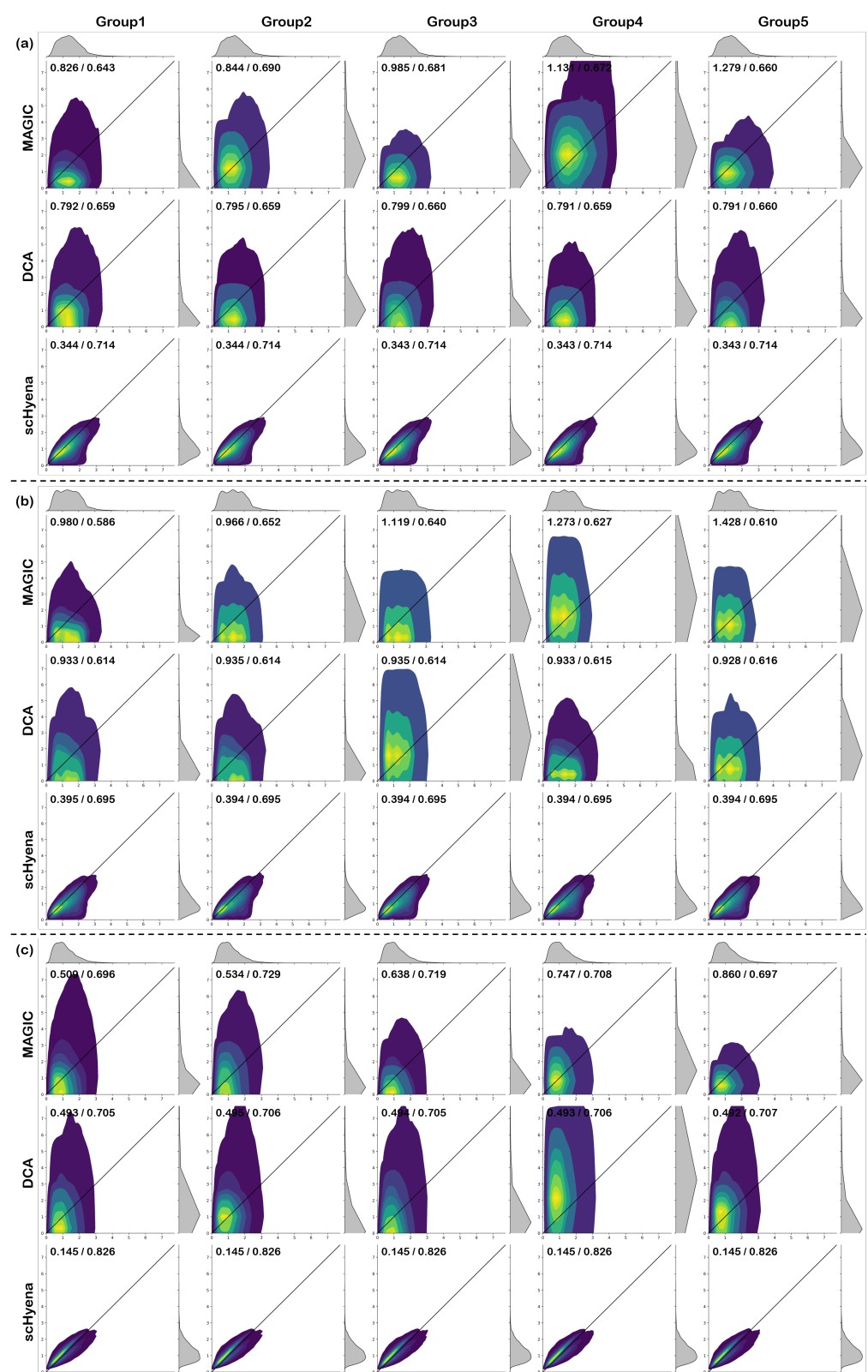

Figure 13: Joint plots comparing true values and imputed values predicted by the imputation methods on the (a) *Lau* dataset (b) *Leng* dataset, and (c) *Zhu* dataset (x-axis: true values, y-axis: imputed values). The values in the upper left corner of each scatter plot represent MSEs and Pearson correlation coefficients.

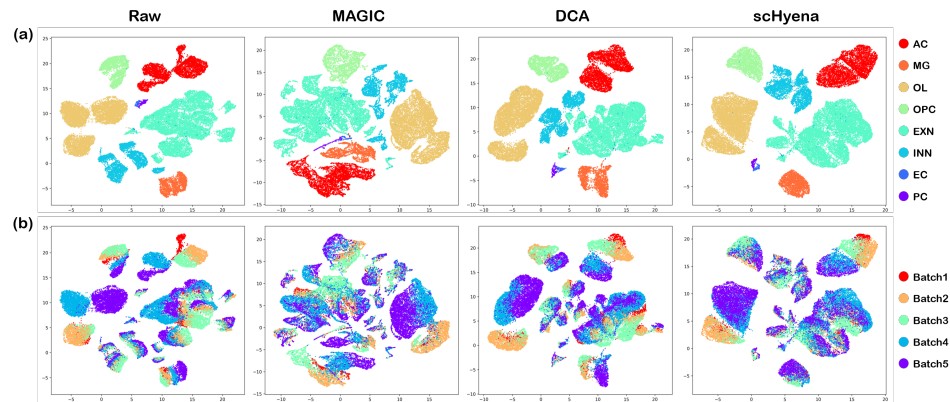

Figure 14: The UMAP visualization of raw and imputed scRNA-seq data for the *Lau* dataset. The figures are labeled with (a) the cell type and (b) batch (patient).

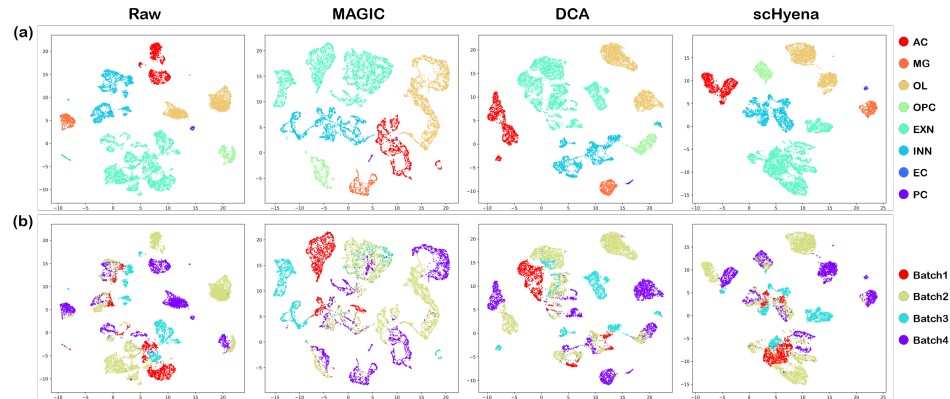

Figure 15: The UMAP visualization of raw and imputed scRNA-seq data for the *Leng* dataset. The figures are labeled with (a) the cell type and (b) batch (patient).

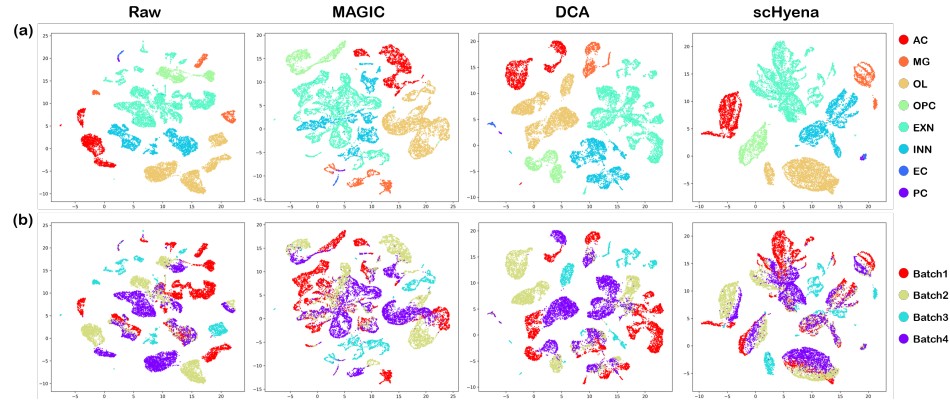

Figure 16: The UMAP visualization of raw and imputed scRNA-seq data for the *Zhu* dataset. The figures are labeled with (a) the cell type and (b) batch (patient).

