# OpenReview forum: "scHyena: Foundation Model for Full-Length Single-Cell RNA-Seq Analysis in Brain"
_ICLR.cc/2024/Conference — ICLR 2024 Conference Withdrawn Submission_

### Official Review · Reviewer_GcHm · 2023-10-28

**Soundness:** 3 good
**Presentation:** 4 excellent
**Contribution:** 2 fair
**Rating:** 5
**Confidence:** 3

**Summary:**

* A single-cell RNA sequencing (scRNA-seq) representation learning method is proposed, utilizing the Hyena operator to replace the self-attention mechanism employed in the transformer-based model. The use of the Hyena operator reduces computational complexity, enabling the processing of scRNA-seq data without the necessity of dimension reduction or the selection of highly variable genes.
* The model is pre-trained on two real datasets and then evaluated on two downstream tasks: cell type classification and scRNA-seq imputation, using two other brain datasets that were not part of the training data.

**Strengths:**

* The model is demonstrated to outperform baseline methods in downstream tasks, with a particularly significant improvement in imputation, as evidenced in the Pearson correlation plots.
* Various techniques have been introduced to adapt to this specific application, including the utilization of bidirectional Hyena, the replacement of discretizing continuous inputs with an adapter layer, and the incorporation of gene embeddings.

**Weaknesses:**

* It's not entirely clear what the primary contributor to the improved performance in cell type classification and imputation is. Is it the use of full-length RNA sequences instead of highly variable genes (HVGs), the pre-training, the model architecture, or the incorporation of embedding techniques? Conducting an ablation study could be helpful in providing insights.
* The primary reason for employing the transformer-based model and the Hyena operator is to learn from the longer sequence length. However, learning from the entire sequence comes with certain caveats, including a large number of model parameters, longer training times, and the risk of overfitting, etc. Since the majority of genes are zeros, it's not clear whether the potential benefits outweigh these drawbacks as described.
* It would be beneficial to provide a comparison of computational complexity in real application, such as runtime and memory usage, especially in comparison to similar methods like scBERT. This is important as one of the main advantages of scHyena is the reduction in computational complexity.

**Questions:**

1. Is scRNA-seq data from brain tissue inherently more challenging to work with compared to data from other tissues, or is there a specific reason for developing the method using brain data?
2. Is the scBERT model correctly employed in the experiment? The results for scBERT appear surprisingly poor, despite sharing many similarities with the proposed model. Could you provide insights into why scHyena significantly outperforms scBERT from a model architecture perspective?
3. Could the author provide further insights into the primary factors contributing to the improved model performance? Specifically, is it the use of full-length RNA sequences instead of highly variable genes (HVGs), the pre-training, the model architecture, or the utilization of embedding techniques?

---

### Official Review · Reviewer_LBHW · 2023-10-31

**Soundness:** 2 fair
**Presentation:** 2 fair
**Contribution:** 2 fair
**Rating:** 3
**Confidence:** 4

**Summary:**

- Hyena operators ([Poli et al. 2023](https://arxiv.org/abs/2302.10866)) are a proposed drop-in replacement for attention. Like attention, Hyena operators enable _global interactions_ among inputs and _data-dependent gating_; however they have a much smaller memory footprint compared to attention.

 - In this manuscript, authors propose a transformer-like model (with Hyena operators swapped-in for attention) for analysis of scRNA-seq datasets in a way that can use all ~20,000 genes as input (rather than the common practice of using smaller subsets of e.g. highly variable genes)

 - The resulting model shows favorable performance on tasks such as expression imputation and classification.

**Strengths:**

- The model is able to use all genes for training
 - Results on imputation look promising

**Weaknesses:**

1. The bidirectional and causal cases of the Hyena operator are both described in [Poli et al. 2023](https://arxiv.org/abs/2302.10866), and therefore should not be claimed as a contribution of this manuscript.

2. Projecting expression values to high dimensional space with a linear operator is claimed as a contribution; however the utility of doing so isn't quite clear and there are no experiments to designed to test this directly.

3. F1 values in Table 1 are all quite high across methods (see also Fig. 12). This suggests the classification task is quite easy. Ground truth annotations are typically noisy, and this could easily explain performance differences across methods.

4. Related to the above, scRNA-seq analyses of brain datasets point to 100's of cell types per brain region in mammals. Recently published results (e.g. [Siletti et al., 2023](https://www.science.org/doi/10.1126/science.add7046)) at the whole brain level suggest the existence of 1000's of cell types in mammalian brains. The classification on 10 somewhat coarse level cell types is a weak test for a proposed foundational model.

5. Claims of correcting batch effects are supported solely by a visual inspection of UMAP plots. Experiments can easily be designed to test batch correction capacity more directly, e.g. by testing on a hold out set of batches. There are also specific

6. Overall, I don't see evidence of substantial technical development. The model performance isn't characterized well nor compared to strong baselines (see recent preprints along these lines: [Boiarsky et al. 2023](https://www.biorxiv.org/content/10.1101/2023.10.19.563100v1.full.pdf) and [Kedzierska et al. 2023](https://www.biorxiv.org/content/10.1101/2023.10.16.561085v1.full.pdf)).

**Questions:**

1. It is unclear how competing methods were trained / tested. For example, given that not all genes are typically used as input to other methods, how are results from these methods obtained/ comparable?

2. Why was imputation was evaluated on 5 groups of genes? How were these groups chosen for different methods?

3. I could not understand Fig 5 at all. Please at least describe in brief what the Wolock et al. 2019 scheme is trying to capture, and how it relates to Fig 5.

4. scRNA-seq data is quite sparse. The number of unique genes expressed in a dataset can be a small fraction of the total number of genes. A rough characterization of this could be helpful to understand what fraction of the ~20,000 genes have any meaningful expression in the datasets considered here.

---

### Official Review · Reviewer_vZHL · 2023-10-31

**Soundness:** 3 good
**Presentation:** 3 good
**Contribution:** 2 fair
**Rating:** 3
**Confidence:** 3

**Summary:**

The authors present a Transformer architecture based on the Hyena operator aiming at analysis of scRNA data coming from complex tissues.

**Strengths:**

Timely and addressing a relevant problem with a new (architecture) approach.

**Weaknesses:**

The main weakness is the evaluation of the proposed approach

**Questions:**

In their title (and paper) they refer to full-length single-cell RNA-seq. There is a difference between the meaning of the term full-length as used in the paper and what is commonly understood in the single cell community. Full-length RNA sequencing usually refers to sequencing that allows for isoform detection, etc. (see https://doi.org/10.1038/s41596-021-00523-3). Here, the authors mean that their model does not require to subset highly variable genes, i.e. their model works with ~19.000 genes.

In their abstract they write: "we introduce scHyena, a foundation model designed to address these challenges and enhance the accuracy of scRNA-seq analysis in the brain." referring to the complexity of the brain and the many different types of cells. However, in the cell type classification task, they use only 8 cell types. To make a more convincing point the authors should consider increasing the granularity of cell types labels in their data. For example the Human Allen Brain Atlas, catalogs 128 cell types only for the primary motor cortex, see https://knowledge.brain-map.org/celltypes/CCN201912131.

In their cell type classification task, they use scrublet to call doublets, which they subsequently use as "ground truth" to show how their model is better at classifying doublets compared to DoubletFinder. It only shows that scHyena is able to mimic scrublet. This is not very convincing and requires further clarification if scrublet is to be considered as a ground truth. See a recent benchmark https://www.sciencedirect.com/science/article/pii/S2405471220304592 showing that it is not entirely clear what method works better. One might claim that DoubletFIndeer might even perform better in some contexts.

What is the reason for training the model with a relatively small number of cells (~400k), whereas scGPT is based on ~10M, and Geneformer on ~ 30M cells?

---

### Official Review · Reviewer_3TUJ · 2023-11-05

**Soundness:** 2 fair
**Presentation:** 2 fair
**Contribution:** 1 poor
**Rating:** 3
**Confidence:** 5

**Summary:**

The paper applied the existing work HYENA into handling long sequence for single-cell data.
As claimed in the paper, it is specifically design for brain tissue, and evaluate in cell type annotation and imputation task.

**Strengths:**

1.	It is good try to apply HYENA into handling long sequence for single-cell data.

**Weaknesses:**

1.	The title should be corrected. If the model is only pretrained for the brain, it is supposed to be called as Foundation model. Foundation model should be a universal model for all tissues or at least most of tissues.
2.	Novelty in this paper is limited to me. Also the author mentions the “dropout” problem for single-cell, I totally I agree that. But the paper doesn’t mention whether or how the proposed solution resolve it.
3.	It is not clear how many cells for the brain used in pretraining.
4.	The downstream tasks should include much more, and only cell type annotation and imputation is not enough to evaluate a foundation comprehensively. The more downstream task can refer to geneFormer, scFoudation, scGPT, CellPLM and so on.
5.	The baselines used to compare with the proposed method is very outdated. More foundation models comparison should be considered, like geneFormer , scFoudation, scGPT, CellPLM.

**Questions:**

1.	How do you resolve the dropout problem in your proposed method?
2.	How about result comparison with other state-of-art foundation models for single-cell? More downstream tasks should be included.
3.	In addition to the capability of handling long sequence data, what’s other benefits from proposed method?